# Contribution of Mangrove Forest to the Livelihood of Local Communities in Ayeyarwaddy Region, Myanmar

**Wai Nyein Aye [1]**, **Yali Wen [1],***, **Kim Marin [1]**, **Shivaraj Thapa [1]** and **Aung W. Tun [2]**

[1]    School of Economics and Management, Beijing Forestry University, Beijing 100083, China;
      wainyeinaye@yahoo.com (W.N.A.); marinkim310@yahoo.com (K.M.); shivaraj_thapa@outlook.com (S.T.)

[2]    Environmental Conservation Department, Ministry of Natural Resources and Environmental Conservation,
      Dawei 14011, Myanmar; aungwunnnatunwunna@gmail.com

*    Correspondence: wenyali2018@bjfu.edu.cn; Tel.: +86-106-233-8455

**Abstract:** Myanmar's forests are socially and economically significant to the country because over 70% of the country's population depends on natural resources for daily needs. We conducted this study with the aim of assessing the extent to which direct and indirect (tangible) benefits of mangrove forest contribute to local livelihoods in the Ayeyarwaddy Region, Myanmar. We used a questionnaire survey ($n$ = 185 households), interview and group discussion for data collection. The study shows that 43% of total household income is generated through selling of forest products collected from the mangrove forest such as firewood, fishes, crabs and prawn, whereas agricultural and non-farm incomes were found to be 25% and 32% of total income, respectively. The result prevails that income from the mangrove forest products for fish, crab, prawn and firewood is specifically 36%, 28%, 9% and 27%, respectively. Hence, we confirmed that local livelihood mainly depends on the mangrove forest ecosystem.

**Keywords:** mangrove forest; local communities; Ayeyarwaddy region; Myanmar; economic; livelihoods

## 1. Introduction

According to the World Alas of Mangrove, Myanmar is the seventh largest mangrove area covering 3.3% of the world's landmass [1]. Mangroves cover an estimated area of 4629.64 sq km [2] making Myanmar the fourth largest mangrove coverage in Asia after Malaysia, Bangladesh and Papua New Guinea [3]. Myanmar shares common maritime boundaries in the Bay of Bengal with Bangladesh, India and Thailand. The continental shelf covers approximately 230,000 sq km with a relatively wider portion in the central and southern parts. The most extensive mangroves thrive in the Ayeyarwaddy Delta, the Thanintharyi Coastline and the Rakhine Coastline. Mangroves in Myanmar extensively grow throughout the coastal strip of the country, providing ecosystem goods and services to coastal communities as well as all other parts of the country. Mangrove forest in Myanmar is rich in biodiversity. It has 34 mangrove tree species out of the global total of around 70 mangrove tree species. Of the total Myanmar primary mangroves, the majority is located on Ayeyarwady floodplains, with the remainder in Tanintharyi and a lesser portion in the Rakhine area.

Similarly, mangroves provide shelter and nursery habitat to the aquatic animals. Win [4] stated that the importance of mangrove to fisheries is apparent especially for the white (banana) shrimp (*Penaeus merguiensis*) which is the most important shrimp species in Myanmar. It depends on mangrove forests for shelter during its juvenile stage. Some species such as tiger prawn (*Penaeus monodon*), *Penaeus indicus* and *Metapenaeus spp* also depend on mangroves at certain phases of their life cycle and the larvae, post larvae and juveniles of some penaeids species enter the estuarine mangrove areas in

Myanmar [4]. Over 90% of marine species were found in the mangroves during some parts of their life cycles which shows a positive correlation between mangrove area and aquatic animals [5].

The majority of poor people in developing countries rely on forests and woodlands for their livelihood because of low income and lack of other alternative means to support their subsistence [6]. While the contribution of environmental goods and services to rural livelihoods are widely documented [7,8], their significance within forest-dependent communities remains insufficiently explored. It contributes significantly to the local economy of the people living around the mangrove forests as well as people living far from it.

The term "nutraceutical" is the combination of "nutrition" and "pharmaceutical" and was introduced by Stephen DeFelice in 1989 [9]. Mangroves are important natural resources that are able to provide a wide range of goods and services for the local community. Further, chemical compounds and extracts of mangroves can be used mainly for folk medicine [10]. Rhizophora seedlings are able to cure a sore mouth [10]. The bark extract of *Brugueria sexangula* (Lour.) Poir. is effective against two tumors of Sarcoma 180 and Lewis Lung Carcinoma [11]. Extracts from the bark of *Rhizophora mucronata* Lamk. and the leaf of *Brugueria cylindricall* (Linn.) show antiviral activity against all the viruses tested [11]. And extracts from the leaves, barks, stems and roots of *Ceriops tagal* (Perr.) C.B.Rob., *Ceriops decandra* (Griffith) Ding Hou, *Xylocarpus granatum* Koen. (Meliaceae), *Xylocarpus moluccensis* (Lam.) M. Roem., *Rhizophora mucronata* and *Rhizophora apiculata* Blume. have shown to have antistringent, antdiarrhoea and haemostatic properties [11]. Extracts from the mangroves have been applied in the treatment of health disorders for centuries. Furthermore, in coastal areas, land is a scarce resource for the local community to fulfill food demands. So, interests have been emphasized on the utilization potential of mangroves. For example, in parts of Papua New Guinea, seedlings of Bruguiera species are the staple food [12] and propagule of *Bruguiera sexangula* (Lour.) is able to be eaten after peeling, soaking and boiling [13]. Priya and Niranjana mention that *B. gymnorrhiza L.* (Lamk) and *B. cylindrica L.* (Blume) are rich in nutritional value such as calcium, iron and magnesium and should be considered famine foods in the coastal areas [12].

Mangrove forests can provide a wide range of tangible and intangible benefits such as clean, safe and healthy environments, many forest products and a wide variety of seafood. Mangrove ecosystem services are worth an estimated US$ 33–57 thousand per hectare per year to the national economies of developing countries with mangroves [14]. Vo et al. [15] confirmed that both goods and services provided by mangrove ecosystems contribute to human well-being directly and indirectly. Similarly, Andy et al. [16] supported that knowing the economic value of ecosystem services is an important asset because a major demand is to support human well-being, sustainability and distributional fairness. If there was no mangrove forest, people who rely on the mangroves would suffer from a lack of forest products and food security, especially in fisheries, reduced crop yield and the direct impact of natural disasters. Therefore, many development factors would be negatively affected by the loss of a mangrove forest.

Concerning the economic value of mangrove forests in Myanmar, Wai [17] conducted a study on economic dependency of local communities on mangroves: a case study in Bogalay, Myanmar. Compared to other areas in the country, mangrove depletion and degradation rate is relatively greater in the Ayeyarwady region of Myanmar due to the higher population, easier access to the forest, conversion to salt-producing land and the devastating impacts of Cyclone Nargis. In Myanmar, since the past three decades, over 58% of mangroves have been undergoing over-exploitation, illegal felling, agricultural expansion and conversion to fishponds and shrimp ponds [18]. Mangrove coverage estimated in 2010 has significantly decreased in the past three decades. Major sources of livelihood activities in that area include paddy cultivation, livestock raising, small- and medium-scale agricultural and fish processing, small-scale forest activities (firewood, charcoal production and timber extraction) and salt production. Most of the livelihoods in that area are not sustainable. To give them a sustainable livelihood, alternatives should be provided for the habitat.

Particularly, the degradation of mangroves in the Ayeyarwady Region is due to extremely high demand of fuelwood for Yangon and cities in adjacent areas. Given a lower population and the fact that the coastal landscape is more sheltered in Rakhine State and the Tanintharyi Region, mangroves are in better condition [19] and there is an increasing need of fuelwood from Yangon city to meet an annual demand of 700,000 tons [20]. In addition to household consumption, fuelwood and charcoal are also supplied to cottage industries, restaurants and tea shops [18]. Cultivation of paddy fields is also another main threat of mangrove conversion to other land use though soil condition is not suitable for agriculture. Agricultural expansion into mangrove areas to meet the requirements of regional food security is also common in the other two coastal regions, especially in the Rakhine region [20].

Moreover, a worrying trend for mangroves in that area is the conversion of mangrove forests into shrimp ponds and agricultural lands as well as to other uses such as salt production. Another serious issue is the extraction of trees for fuelwood and charcoal making. As a consequence of many drivers, mangrove deforestation in the Ayeyarwaddy region is recognized as a critical environmental issue for the country. The Ayeyarwady region is the most populated state of Myanmar where 88% of people live in rural areas. Local people, particularly landless labors, generate their subsistence and income from the mangrove forest through the collection of firewood, production of charcoal, harvesting of fisheries and collection of material for shelter. Furthermore, this area was seriously affected by Cyclone Nargis in 2008 and rural communities living around the mangrove area heavily depend on mangrove products directly as well as indirectly either for subsistence use or commercial purposes. Thus, the quantification of mangrove forest contribution to rural livelihood is important for the conservation of this area.

The study was conducted to assess the extent to which direct and indirect (tangible) benefits of mangrove forest resources contribute to the livelihoods of adjacent communities in the Ayeyarwaddy Region in Myanmar. This study seeks to answer the following questions: (1) What are the mangrove forest products that the local communities receive from mangrove forest? (2) How to access these mangrove forest products for the livelihoods, directly or indirectly? (3) How much will daily income generated from the livelihood system be? (4) What are the major income sources in that area? (5) How important is mangrove forest for the livelihood of the local community?

## 2. Materials and Methods

### 2.1. Study Area

The research was carried out in the western part of Meinmahla Kyun Wildlife Sanctuary, Bogalay Township, Ayeyarwaddy Region of Myanmar. The location of Meinmahla Kyun Wildlife Sanctuary is 15° 57.822′ N and 95° 17.988′ E. This wetland reserve is on Meinmahla Kyun and is classified as a mangrove reserve. This reserve area has 136.72 sq km and was established in 1986. It is the third Ramsar site of Myanmar and was designated in 2017. It is a coastal wetland in the southern part of the Ayeyarwaddy Delta which is also an ASEAN Heritage Park. It supports one of the largest remaining mangrove areas of the Delta where mangrove ecosystems have been declining due to activities including logging, charcoal and firewood production, fishing and development of shipping lanes. At present, the mangrove species are being replaced by mangrove date palm (*Phoenix paludosa* Roxb.). It supports globally threatened species such as hawksbill turtle (*Eretmochelys imbricate*), mangrove terrapin (*Batagur basaka*), the endangered great knot (*Calidris tenuirostris*), Nordmann's greenshank (*Tringa guttifer*), green turtle (*Chelonia mydas*), dhole (*Cuon alpinus*) and vulnerable species of the Pacific ridley turtle (*Lepidochelys olivacea*), fishing cat (*Prionailurus viverrinus*), lesser adjutant (*Leptoptilos javanicus*) and the Irrawaddy dolphin (*Orcaella brevirostris*) [18]. The site is also the last estuarine habitat in Myanmar for the saltwater crocodile (*Crocodylus porosus*). It holds significant cultural and historic value for the people of Myanmar based on myths and pilgrimages which closely connect them to their environment [21].

Two villages—Padekaw village (15° 59.232′ N and 95° 15.765′ E) and Lawinekyun (A Nauk) (16° 0.586′ N and 95° 15.866′ E)—were selected as study areas to analyze the socio-economic conditions.

These two villages were selected with the criteria of accessibility, near the Meinmahla Kyun Wildlife reserve, and affected by cyclone Nargis in 2008. Because most of the villages in that area can only be accessed by boat, we selected these two research areas as they are a little easier to access by boat. Table 1 shows the general information of the two villages. Local communities living in that area survive by working cultivation of paddy fields and fisheries. Sources of employment include crop farming (mainly paddy rice cultivation), horticulture (mostly fruit trees), paid agricultural labor, fishing (fishponds, shrimp farms, inland and offshore fisheries), small- and medium-scale agricultural and fish processing and small-scale forestry activities (firewood, charcoal and timber). Some income is derived from commerce and small-scale local trade which is indirectly reliant on the environment as the target customers of those trades and businesses are natural resource-dependent. Most of the economic activities in that area are at the subsistence level. Most men in the coastal areas are fishermen while women and children are collectors of inter-tidal mollusks, fish and prawns. These products are an important source of income to both fishermen and those engaged in processing and trading. The Delta region has many challenges such as capacity development and infrastructure for hygienic drinking water and better education and healthcare. Most of the households use firewood as fuel energy for cooking and heating. Access to the villages of the studied area was difficult because of poor transport infrastructure and few all-season roads that travel between villages that are often conducted by boat. Households in the studied villages use rainwater for drinking and cooking, however they often cannot collect enough rainwater. Figure 1 shows the location of the study area.

**Table 1.** General information of the two villages.

| Village Name | Township Name | Total Households |
|---|---|---|
| Padekaw | Bogalay | 245 |
| Lawinekyun (A Nauk) | Bogalay | 100 |
| **Total sampling households** | | **345** |

La Waing Kyun(Ah Nauk) and Pe De Kaw Villages, Ayeyarwady Region

**Figure 1.** Map of the study area showing La Waing Kyun, Pa De Kaw Villages and the Ayeyarwady Region.

*2.2. Framework of the Study*

This study is based on the sustainable livelihood framework (SLF) for the analysis and assets. In SLF analysis, the relative importance of five types of assets (natural, physical financial, social and human) is evaluated. These assets (both material and social resources) constitute means that households use in the pursuit of their livelihood strategies. Factors that influence the mangrove forest

income were selected based on the five that were assessed for sustainable livelihoods. In SLF, wealth, age, gender and skills are considered as financial, social and human assets.

We selected 185 sample respondents as a sample size (at 95% confidence interval and 5% marginal error) for this study by using Taro Yamane Formula [22].

$$n = \frac{N}{(1 + Ne^2)} \tag{1}$$

where n = sample size; $N$ = total population of household; $e$ = allowable error (5% = 0.05).

*2.3. Methodology*

This section contains the discussion on the method of data collection and data analysis that was used in this study.

The introduction of the research was guided to know the general information and socio-economic condition of the research area. Then, the survey was conducted during the periods of August and September 2018. For this research, a cross-sectional research design was used because this method is suitable for time and financial limitations. This design is also accurate and provides quick results. This study mainly relied on primary data and data was collected by face to face interviews. For an interview, well-prepared structured questionnaires (both open-ended and close-ended) were used. The local dwellers living and depending directly and indirectly on mangrove forest products for their daily livelihood system were selected as respondents for the interview survey. The simple random sampling method was used in selecting a sample household for the interview. A total of 185 (~185.24) households with 95% precision level were used as the sample size by calculating Taro Yamane Formula [22].

The main focus of this study was to analyze the contribution of mangrove forest products' income on the livelihood of local dwellers. To show the link between the contribution effect of mangrove forest resources and its impact on livelihoods, education, household composition, age, land and sources of family income, multiple linear regression models were used. The multiple linear regression model was estimated by using the ordinary least squares estimation technique (OLS) after the data was checked for different econometric tests. We used OLS multiple regressions to build models of the household characteristics associated with mangrove forest product earnings. The income generated from mangrove forest resources was regressed as a function of other socio-economic characters; the economic model was as follows.

$$Y = \beta_0 + \beta_1 x_i + u, \tag{2}$$

where Y = the income from the mangrove forest; $\beta_0$ = intercept, $\beta_1$ = estimated coefficients of the explanatory variable; $x_i$; $x_i$ = explanatory variables (socio-economic characteristics); u = error term.

Total household income was estimated as follows:

$$\text{Total Household income} = \Sigma \text{ (income from agriculture + Non-farm income + Income from mangrove forest)} \tag{3}$$

In this study, income was calculated in the currency of Myanmar (Kyat). According to the exchange rate by the central bank of Myanmar (2018), US$ 1 is equal to about 1428.6 Kyats. Major income sources are agriculture, non-farm activities and collection of mangrove forest products. Agricultural income includes income from the cultivation of crops for purposes of both household consumption and selling. Information on crop yields was gathered from a household respondent through the questionnaire survey. Prices of crops were obtained from the local market. For non-farm income, it includes all income from wage labor, employment such as government staff and private shops. Wage labor in the study area was mostly in a mangrove forest plantation. The daily wages for men and women were not the same. The wage rate and the number of working days/hours reported by the respondents were used for the estimation. Income from private shops was obtained from the individual household respondent through the interview. And the final income source is income from the collection of mangrove forest

products. Information about the collection and sale of mangrove forest products (firewood, shrimp, fish and crab) was obtained from the household questionnaire. In addition, data regarding different kinds of mangrove forest products and their price was obtained using the key informant survey and the questionnaire interviews. Monthly income was gathered from respondents through questionnaire and it was converted into annual values.

Both qualitative and quantitative techniques were used for analyzing data. Before processing the responses, the completed questionnaires were revised for completeness and consistency. Qualitative data were summarized by way of text analyses, while quantitative data were analyzed by descriptive statistics and OLS regression analysis. Descriptive statistics such as frequencies, percentages, mean value and standard deviation were computed for all the quantitative variables and information and were presented in the form of tables and graphs. Descriptive statistics were used because they enabled the research to meaningfully describe a distribution of scores or measurements using a few indices. The collected data was classified, tabulated and analyzed in Microsoft Excel and STATA version-13.

## 3. Results

### 3.1. Basic Characteristics of Sample Households and Annual Household Income

A simple random sampling survey was conducted in the study sites with prepared questionnaires during the two-month period from August to September 2018. A total of 185 sample households were interviewed from two villages of Bogalay Township, Ayeyarwaddy Region of Myanmar. Table 2 shows a descriptive analysis of household respondents. According to the results of the study, the gender distribution of the household head shows that 91.89% are male and the remaining 8.11% are female. Regarding the age distribution of respondents, only 1% of the respondents are 19 or less than 19 years old, 19% of the respondents are aged between 20 and 29 years, 27% of the respondents are aged between 30 and 39 years, 25% of the respondents are aged between 40 and 49 years and the remaining 28% of the respondents are above 49 years old.

Family size varied from 1 to 10 members with a mean value of 4.14 (standard deviation, sd = 1.65). In terms of education levels, 1% of respondents were graduates, 3% had attended high school, 22% studied at secondary school, 30% studied up to primary school and 44% of respondents had traditional Buddhist monastic education. So, this means that most of the respondents did not have a formal education. For old people, they had access to traditional monastic education. Crop farming (mainly paddy rice cultivation) is a major source of livelihood in Myanmar. The Ayeyarwaddy region is also well known as the "rice bowl of Myanmar". Among the respondents, however, 26% of households owned agricultural land and the remaining households were agriculturally landless. Minimum and maximum agricultural land holding sizes of respondents were 0 and 100 acres, respectively, with a mean value of 4.33 (standard deviation of 10.90). In the research area, 77% of households derived their income from various sources such as causal and seasonal labor in agriculture, wage labor in mangrove forest plantations, small scale trade, shop keeping, collection of firewood, fish proceeding and crafts. Then, 83% of households were non-native villagers. Further, 17% of respondent's houses were made of metal roofing and timber flooring, 8% were made of metal roofing, brick and concrete and wood, 74% of houses were made of Nypa roof, timber and bamboo flooring and the remaining 1% do not own a house and live on a boat. Some respondents harvested construction materials for their houses from the forest. For example, nipa palms (*Nypa fruticans*) along riversides are over-harvested for thatching, while Palmyra species on riverbanks and around paddy fields are utilized for both thatching and timber.

The sources of income in the study sites are farm activities, non-farm activities and collection of mangrove forest products. Among the respondents, 53% of the respondents generated their income from the collection of mangrove forest products, 23% from agricultural activities and the remaining 24% from non-farm activities. According to the results of the Table 3, mangrove income makes up 43% of the

total household income. It included both subsistence and cash income. Agricultural income shares 25% of the total household income and non-farm income accounts for 32% of the total household income.

**Table 2.** Descriptive analysis of household respondents.

| Household Characteristics | | Frequency | Percentage (%) |
|---|---|---|---|
| Age (years old) | 19 or Less than 19 | 2 | 1 |
| | 20–29 | 36 | 19 |
| | 30–39 | 50 | 27 |
| | 40–49 | 47 | 25 |
| | More than 49 | 50 | 27 |
| (Minimum = 19; Mean = 41; Maximum = 72) | | | |
| Gender | Male | 15 | 8 |
| | Female | 170 | 92 |
| Education | Graduate | 1 | 1 |
| | High School | 5 | 3 |
| | Secondary School | 41 | 22 |
| | Primary School | 56 | 30 |
| | Non formal Education | 82 | 44 |
| Occupation | Farmer | 43 | 23 |
| | Non-farm activity | 98 | 53 |
| | Collection of mangrove based products | 44 | 24 |
| Access to mangrove products | Directly | 121 | 65 |
| | Indirectly | 64 | 35 |
| Native village | Yes | 154 | 83 |
| | No | 31 | 17 |
| Agricultural land | Yes | 49 | 26 |
| | No | 136 | 74 |
| (Minimum land size = 0, Mean land size = 4.33, Maximum land size = 100) | | | |

**Table 3.** Average annual household income.

| Types of Income | Average Income Per Year (Kyats/year/household) | Standard Deviation | Standard Error | Income Share (%) |
|---|---|---|---|---|
| Mangrove Forest Income | 1,119,957 | 1,659,339 | 121,997 | 43% |
| Agricultural Income | 642,573 | 1,521,448 | 111,859 | 25% |
| Non-Farm Income | 833,157 | 2,985,505 | 219,499 | 32% |
| **Total Income** | 2,595,687 | 3,365,090 | 247,406 | 100% |

*3.2. The Contribution of Major Mangrove Forest Products*

Income from the collection of mangrove forest products was the highest income source in the study. About 53% of sample households generated their livelihood income from the use of different mangrove forest products. The result indicates that mangrove products harvested by the local people from the study area are fuelwood, fish, crab and prawn. The main contributions in both villages are fish and crab. Their usage of timber products collected from mangrove forest is mostly for subsistence purposes such as building material and firewood because in that area they mainly use firewood as fuel for cooking. These two studied villages are dependent on the mangrove forest resources provided by a Meinmahla Kyun wetland reserve. Timbers from mangrove trees are used as poles, firewood and charcoal making for domestic purposes such as cooking, heating and ironing. Mangroves are the source of fuelwood for cooking in the rural area. Fisheries and prawn catch in particular depend on intact mangrove ecosystems. Villagers in the studied sites engage in traditional fish collection from the mangrove areas. The mangrove dwellers basically understand the daily tidal conditions by calculating the days based on the Myanmar lunar calendar [23]. Almost all of the respondents in two villages collected fish, crab and prawns at 10 to 15 days per month as they were dependent on a tidal cycle. Tidal inundation is more frequent and widespread during the period of the lunar cycle. During that

period, people could harvest more fish, crab and prawn. Income share from fishing, crab, prawn and firewood were 36%, 28%, 9% and 27%, respectively. Figure 2 presents the major mangrove forest products in the study area.

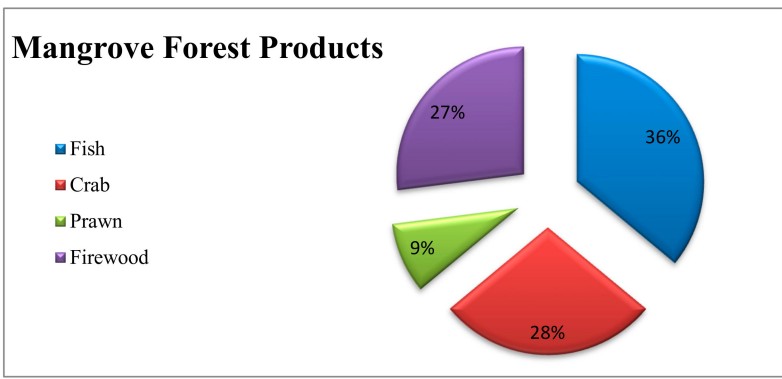

**Figure 2.** Major mangrove forest products in the studied area.

### 3.3. Mangrove Forest Income among Different Income Level

According to the results obtained as shown in Table 4, we can see clearly that the income from the mangrove product is higher at the low and middle level. To differentiate households in the form of wellbeing, we separated households based on their total income as; high, medium and low income level households. Most of the local poor people in that study area were landless and they do not have a regular income. So, in this study, it was decided that income level of less than US$ 1000 per year constitutes being poor, an income level of US$ 1000–1700 per year is medium and an income level of more than US$ 1700 per year is the high level. The income sharing from mangrove products in the high-income level was 32.8% and in the middle-income level was about 52.8%. Local communities with lower income levels generated the most mangrove products which contributes 79.4% to the total income. Most of the middle-income and low-income level households were landless and are absolutely dependent on mangrove forest resources for their livelihood activities. So, farm incomes at low- and middle-income levels were 9.4% and 10.5%, respectively. At the highest income level, households own agricultural land and have better off-farm jobs such as a private shop. Farm income and non-farm income sharing at a high level were 33.3% and 33.9% of the total income, respectively. This means that forest dependency will reduce if there are other better alternatives.

**Table 4.** Income sharing among different income levels.

| Income Source | High Income (*n* = 56) | | Medium Income (*n* = 65) | | Low Income (*n* = 64) | | Kruskal-Wallis Test |
|---|---|---|---|---|---|---|---|
| | Income | % | Income | % | Income | % | |
| Mangrove Product income | 1,791,000 | 32.8 | 956,862 | 52.8 | 698,438 | 79.4 | *p* = 0.1930 X2 = 3.290, df = 2 |
| Agricultural Income | 1,819,571 | 33.3 | 169,692 | 9.4 | 92,969 | 10.5 | *p* = 0.0001 X2 = 33.753, df = 2 |
| Non-Farm Income | 1,855,179 | 33.9 | 685,569 | 37.8 | 88,782 | 10.1 | *p* =0.0001, X2 = 19.482, df = 2 |

Note: 1US$ = 1428 MM Kyats in 2018.

### 3.4. Mangrove Forest Income against Socio-Economic Characteristics

The relation of socioeconomic characters of the respondents and mangrove forest resources use was addressed by using a multiple linear regression model which showed mangrove forest income was regressed as a function of different socio-economic characters (factors that affect the level of

forest dependency) of the respondents such as agricultural land size, household size, ways to access mangrove forest products, native village, occupation, age, gender of household head and education. The multiple linear regression model was estimated using ordinary least square estimation technique (OLS) after the data was checked for different econometric tests.

The regression result is shown in the Table 5 and many explanatory variables have expected influence on forest dependency. The model explained 35% of the variance on mangrove forest income (F = 10.68, $p < 0.000$). While coefficients on the agricultural land size, the way of accessing mangrove forest products, the occupation of farm activities were statistically significant at (1%), variable, household size was statistically significant at (10%) and variable, non-farm activities was statistically significant at (5%).

This showed that mangrove forest income was negatively correlated with native village, education and occupation activities such as farm activities and non-farm activities. On the other hand, agricultural land size, household size, the ways of getting mangrove forest product, age and household head are positively related to mangrove forest income. However, the variables such as gender of household head, age, education and native village are not statistically significant at any level of significance which shows that those variables were the least important determinants of the household dependence on forest resources.

**Table 5.** Ordinary least squares estimation (OLS) regression of mangrove forest income against household characteristics. Number of obs = 185; F (8,176) = 10.68; Prob > F = 0.0000; R-squared = 0.3545; Adj R-squared = 0.3213.

| Variable | Estimated Coefficient | T Ratio | $p > (t)$ |
|---|---|---|---|
| Agricultural land size | 87,001 *** (12,945) | 6.72 | 0.000 |
| Household size | 109,253 * (63,621) | 1.72 | 0.088 |
| Way of accessing mangrove products | $1.1 \times 10^6$ *** (305,294) | 3.60 | 0.000 |
| Native village | −310,169 (281,029) | −1.10 | 0.271 |
| Non-farm activities | −671,806 ** (331,192) | −2.03 | 0.044 |
| Farm activities | $-1.857 \times 10^6$ *** (385,256) | −4.82 | 0.000 |
| Age | 27,057 (101,631) | 0.27 | 0.790 |
| Gender of HH Head | 181,432 (403,874) | 0.45 | 0.654 |
| Education | −141,803 (125,741) | −1.13 | 0.261 |
| Constant | 705,632 (692,604) | 1.02 | 0.310 |

Notes: Standard errors in parentheses, *** $p < 0.01$, ** $p < 0.05$, * $p < 0.1$.

## 4. Discussion

### 4.1. Dependence on Mangrove Forest Resources

In the study sites, the majority of the people's livelihoods were at subsistence level. They heavily depended on natural resources for their livelihoods. Major livelihood activities in the study sites were agriculture, non-farm activities and mangrove forest product collection. Among them, mangrove

forest resources were the major income source and most of the coastal communities have relied on them. The main provisioning services of mangroves are timber, charcoal and firewood as energy sources, shelter, fodder, medicines and a fishery which is important for subsistence, livelihood and commercial fisheries for the communities living in coastal and delta areas. The income for the local poor communities living in rural area of the developing countries was less than US$ 1 per day and they rely on the ecosystem services-ES [24]. Their income (43% to total household income) were generated by selling forest products collected from the mangrove forest such as fishes, crabs and prawn. So, half of the respondents were engaged in mangrove based occupations because they are poor and predominantly live in the delta region. The average annual household income from mangrove forest products per year was Kyats 1,119,957 (approximately US$ 784). Wai [17] stated that the economic value of the mangrove was USD$ 1497.6 (approximated Kyats 2,139,471) in her research of "Economic Dependency of Local Communities on Mangroves: A Case Study in Bogalay, Myanmar". This means that the economic value of mangrove forests is gradually decreasing due to the deforestation and degrading of mangrove forest. Levels of dependence on forest resources around the world among households with access to forests vary from 6 to 65% depending on the local circumstances [25–42]. Singh [43] in Bangladesh estimated that the contribution of non-timber forest products-NTFPs is 79% on average to the annual income of the collector's family. Clinton, U.I and Okujagu, C.M.D [44] inferred that in their study, 85% of households depended on mangrove resources for their income. In this study, agricultural income estimated about 25% of total income. Paddy field is the major cultivation in the study sites. Seaweed cultivation has rapidly emerged as another cash crop in the coastal area; women were mainly involved in seaweed cultivation. Non-farm incomes accounted 32% of total household income. Major non-farm activities were wage labor in mangrove forest plantation, causal and seasonal labor in agriculture, salary, private shop, etc. Furthermore, mangrove forest dependencies vary among different income levels. According to the result, households with middle-income and low-income levels are the most dependent on forest resources with 52.8% and 79.4% of total household income because most of the middle income and low-income level households are landless and they do not have other alternative income activities. This finding was similar to the finding of Abu Nasar Mohammad Abdullah [45] wherein lower income households were relatively more dependent on forest incomes than the better off households.

*4.2. Factors Influencing Forest Dependency*

The mangrove forest dependence level of rural households was calculated using the relative forest income as a share of total household income account derived from the consumption and sale of mangrove forest resources. The level of dependence (the ratio of mangrove forest income from the total household income) was 43% in the study area on average. So, local households in the studied areas are mainly dependent on the forest resources for their livelihood activities. In this research, socio-economic characteristics that influenced forest dependency were also explained. Agricultural land size is positively correlated with mangrove forest income. This result is contradictory to the general findings of other studies. Lebmeister et al. [46] observed that NTFPs dependency in the rural household was significantly decreased with increasing farmland. In Ethiopia, the relative income from the forest was negatively correlated with cropland [34]. In parts of the Ayeyarwady Delta, land degradation and declining soil fertility due to exploitative farming practices have contributed to decreasing agricultural yields. As a result, in order to maintain agricultural incomes and food production, farmers have resorted to cultivating even more land [47]. For instance, in coastal areas, converting mangrove areas to rice farms has resulted in seawater encroachment and salinization of soils, providing a source of income for only a short period of time before yields drop below economic levels [47]. Household size is directly related to forest income. As the household size increased, the dependency on mangrove forest resources of the household also increased. Ways of accessing mangrove forest products are the main determinant of being dependent on mangrove forest products. According to the result of the survey, 65% of respondents produced mangrove forest products directly and the remaining 35% produced

indirectly. Education level in this study negatively impacted mangrove forest dependence because they have less access to alternative income sources. Mulatie Chanie and Tesfaye Yirsaw [48] also found in the study that education level has a negative impact on the forest dependence of an individual. This means that forest income of the non-educated household is greater than the educated one and shows that a household with educated members is less dependent on forest resources as a means of livelihood income. In this study, most respondents were extremely dependent on the forest regardless of the gender of the head of the household, a similar to the finding of Abdullah [45]. Similarly, it found a negatively significant correlation with mangrove forest income. So, if the households have other alternative livelihood sources, their dependency on mangrove forest will decrease.

## 5. Conclusions

Income from mangrove forest products, agricultural income and non-farm income are the sources of local people for fulfilling their subsistence needs. However, the local people living nearby mangrove forest reserve depend much more on mangrove forest as they can access the mangrove forest products easily in order to generate their income. Income from mangrove forest products is the main income sources of their livelihood income and generates 43% of the total income of the household income. So, households are significantly dependent on mangrove products. The lower level household income group had neither land for agriculture nor off-farm employment for generating their income, increasing their dependency on the forest resources for survival. People are generating their livelihood income from the use of different mangrove resources like fish, crab, prawn and firewood. Firewood is a source of energy for cooking where some households collect firewood for commercial purposes. The second largest source of income is off-farm income which accounts for 32% of the total livelihood income. And agricultural income shares 25% of household income. Lower and middle income level households are more dependent on mangrove forest products when compared to high income levels. Lower income level groups are normally landless and mostly depend on mangrove forest products for their subsistence.

Mangrove forest resources are a major income contribution in the livelihoods of local communities, although few households engage in other alternative livelihood activities, such as agriculture and non-farm employment.

Mangrove forest resources provide an important contribution to local livelihood, therefore issues on forest resource dependency and subsistence level of rural livelihood should not be ignored in policy level decisions and other interventions. In addition to the forest resource use, other income generation activities should be incorporated so that livelihood strategies can be diversified to sustain local livelihood and reduce their dependency on forest resources. To avoid deforestation and inefficient utilization of forest resources, the government needs to implement alternative income generation and forest rehabilitation activities for the protection, conservation and utilization of mangrove resources. Improvement of mangrove forest condition will assure benefit optimization and sustainable management of forest resources.

**Author Contributions:** For research, W.N.A. collected and organized all the data. The analyses were conducted by W.N.A. and guided by Y.W.; K.M., S.T. and A.W.T. helped with the conceptualization and partly advised in the process of writing the paper. All authors considered the outline and contributed to writing the manuscript.

**Funding:** This research is supported by the Natural Science Foundation of China (71861147001, 71373024).

**Acknowledgments:** I would like to express sincere gratitude to APFNet (Asia Pacific Network for Sustainable Forest Management and Rehabilitation) for supporting me with a scholarship to pursue a master degree in the School of Economics and Management, Beijing Forestry University, China. I wish to express my thanks to the Forest Department of Pyaphon District, Bogalay Township for their kind support during my fieldwork. Finally, thanks go to the Forest Department, Ministry of Natural Resources and Environmental Conservation for giving me a chance to study abroad.

**Conflicts of Interest:** The authors declare no conflicts of interest.

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
