# Peer review of "Contribution of Mangrove Forest to the Livelihood of Local Communities in Ayeyarwaddy Region, Myanmar"

_forests, doi:10.3390/f10050414_

Round 1

Reviewer 1 Report

I found the topic of the paper quite interesting and recognize that the socio-ecological system under study is quite important from both conservation and development perspectives. I believe the authors made a major effort to collect information from households and succeeded in showing that households living in the proximity of mangroves are highly dependent on them for their livelihoods. 

On the other hand, the paper has many aspects that need to be improved. The first issue that becomes apparent relates to language which would greatly benefit with strong support from a native English speaker with strong writing skills. The other issue is more problematic since it relates to how the information collected was analyzed and how the results are presented. I would suggest placing less emphasis on trying to develop precise statistics and use the information collected to tell the story that is unfolding in these important ecosystems. 

Author Response

Point 1: I found the topic of the paper quite interesting and recognize that the socio-ecological system under study is quite important from both conservation and development perspectives. I believe the authors made a major effort to collect information from households and succeeded in showing that households living in the proximity of mangroves are highly dependent on them for their livelihoods.

Response 1: Dear Sir, thank you for your review. We modified the paper according to your comment.

Point 2: In the other hand, the paper has many aspects that need to be improved. The first issue that becomes apparent relates to language which would greatly benefit with strong support from a native English speaker with strong writing skills. The other issue is more problematic since it relates to how the information collected was analyzed and how the results are presented. I would suggest placing less emphasis on trying to develop precise statistics and use the information collected to tell the story that is unfolding in these important ecosystems.

Response 2: We modified the paper.

Reviewer 2 Report

In general, the authors should be commended for doing research in such a remote area on a relatively important topic – contribution of mangrove forests on local livelihoods. It seems in Myanmar it is a very pressing issues. Good things about the study were extensive data collection, and some interesting findings were presented. On the other hand, there are also major flaws. It is important that the authors do not only show the contribution of mangrove forests to livelihoods, but that they also analyze more systematically the underlying factors and drivers. Hence, I would suggest the authors to restructure the paper, to highlight more the underlying factors.The level of English is also problematic, and I strongly suggest the authors to let a native speaker to revise the paper. Some other comments: The authors state “While the contribution of environmental products and services to rural livelihoods are widely documented [6,7], their significance within forest-dependent communities remains insufficiently explored.” -> is this really true? I tend to disagree. A lot of research on this topic has already been done. How were the villages selected? Figure 1 is really too small -> please find another map. How many households were interviewed 185 or 345? How were the households selected? What was the sampling frame of the study? Table 2: Why do you have no data on respondents age, instead of using age classes? What does a mean of 3.58 say? What does occupation, 1,2,3 mean? Why not use frequency test? How were the factors chosen for the regression model? It is very important to explain more why factors were significant and insignificant in the regression model. Does it make sense? How will it add to the literature? It should be done more systematically.

Author Response

Point 1: In general, the authors should be commended for doing research in such a remote area on a relatively important topic – contribution of mangrove forests on local livelihoods. It seems in Myanmar it is a very pressing issues. Good things about the study were extensive data collection, and some interesting findings were presented.

 Response 1: Dear Sir, thank you for your review. We modified the paper according to your comment.

 Point 2: On the other hand, there are also major flaws. It is important that the authors do not only show the contribution of mangrove forests to livelihoods but that they also analyze more systematically the underlying factors and drivers. Hence, I would suggest the authors to restructure the paper, to highlight more the underlying factors. The level of English is also problematic, and I strongly suggest the authors to let a native speaker to revise the paper.

 Response 2: We modified the paper. 

Point 3: Some other comments: The authors state “While the contribution of environmental products and services to rural livelihoods are widely documented [6,7], their significance within forest-dependent communities remains insufficiently explored.” -> is this really true? I tend to disagree. A lot of research on this topic has already been done. How were the villages selected?

 Response 3: That is true. In Myanmar, it was a little literature concerning with the contribution of mangrove forests to the local communities.

Point 4: Figure 1 is really too small -> please find another map. How many households were interviewed 185 or 345?

 Response 4: I changed the map of the study area.

 Point 5: How were the villages? How were the households selected?

 Response 5: These two villages were selected with the criteria of accessibility; near the Meinmahla Kyun Wildlife reserve, and affecting the impact of cyclone Nargis in 2008. Actually, most of the villages in that area can only assess by boat as transportation. So we selected these two research area that is a little easier to access by boat. The simple random sampling method was used to select households. 

Point 6: What was the sampling frame of the study? 

Response 6: I put the sampling frame of the study. The research is based on the sustainable livelihood framework.

Point 7: Table 2: Why do you have no data on respondents age, instead of using age classes? What does a mean of 3.58 say? What does occupation, 1,2,3 mean? Why not use the frequency test?

 Response 7: I edited the table (2).

 Point 8: How were the factors chosen for the regression model? It is very important to explain more why factors were significant and insignificant in the regression model. Does it make sense? How will it add to the literature? It should be done more systematically.

Response 8: I used the concept of the sustainable livelihood framework. 

Reviewer 3 Report

In this paper, based on the qualitative and quantitative information collected from surveys, authors attempted to assess the contribution of mangrove forest to livelihood of local communities. This research might have important policy implications, but several issues should be addressed before publishing it in Forests.

Major comments:

The biggest concern I have is whether we can include fish, crab, and prawn as forest products. It is indeed subjective, but to me they are not direct forest products. Rivers, lakes, streams might be inside the forests in the study villages, but still it is they should be in different category from firewood, pole, any medicinal plants etc. Figure 2 shows that only 27% of the mangrove forest income is from fuelwood. Authors should clearly address it with supporting evidences and references.

If readers just look at Table 2, it has all incomplete information. The way data are presented, it reads like those household characteristics are quantities (not categorical or dummy variables). Authors should consider adding description of the variables in a separate column (how they define these variables), so that it is easier to follow that age variable has five categories etc.

It is also unclear how authors came up with three different income level in Table 4. What are the criteria/thresholds to have high, medium and low income levels of the respondents? More details are needed.

The OLS regression results are also confusing and hard to follow. Where the variable of way of accessing mangrove product came from? It is not introduced anywhere (not in Table 2) before. What does "native village" represent? What are non-farm and farm activities and how these variables are constructed i.e. what do they represent? Both non-farm and farm activities variables have negative signs, what do they mean? And, even if the model has 9 variables, R-squared is low (just .35), any comments/thoughts?

The agricultural land size has a positive effect on mangrove forest income, which I think is not an expected sign. The positive sign means the higher the ag land size, the higher the forest income. How can it be explained? I don't know how mangrove forest and agriculture are linked in the study regions, but mangrove forest income is all about seafood and firewood.

Where are the qualitative information and how they are presented? 

I think the manuscript requires a round of English edit. Some of the sentences make less sense. The very second sentence in Abstract is hard to understand. In some places, I, my (personal pronouns) are used but several authors are included in the authors list---it should be we, ours. 

Similarly, the manuscript could be shortened, particularly the intro section, but more details are needed in survey administration, description of variables and results interpretation. 

Author Response

Point 1: In this paper, based on the qualitative and quantitative information collected from surveys, authors attempted to assess the contribution of mangrove forest to livelihood of local communities. This research might have important policy implications, but several issues should be addressed before publishing it in Forests.

 Response 1: Dear Sir, thank you for your review. We modified the paper according to your comment.

 Point 2: The biggest concern I have is whether we can include fish, crab, and prawn as forest products. It is indeed subjective, but to me they are not direct forest products. Rivers, lakes, streams might be inside the forests in the study villages, but still it is they should be in different category from firewood, pole, any medicinal plants etc. Figure 2 shows that only 27% of the mangrove forest income is from fuelwood. Authors should clearly address it with supporting evidences and references.

 Response 2: The direct use value of mangrove ecosystems is a value generated from the utilization of mangrove resources directly. Direct use value is defined as benefits (goods and services) that can be consumed. In this research, the indicator of direct use is fishes and wood production. Snedaker [1978] also stated that over 90% of marine species were found in the mangroves during some parts of their life cycles. So, it has a positive correlation between the mangrove area and aquatic animals. 

Point 3: If readers just look at Table 2, it has all incomplete information. The way data are presented, it reads like those household characteristics are quantities (not categorical or dummy variables). Authors should consider adding description of the variables in a separate column (how they define these variables), so that it is easier to follow that age variable has five categories etc.

Response 3: We edited table 2.

 Point 4: It is also unclear how authors came up with three different income level in Table 4. What are the criteria/thresholds to have high, medium and low income levels of the respondents? More details are needed.

 Response 4: Income levels were separated based on their total income. Most of the local poor in that study area were landless and they do not have a regular income. So, in this study, it was decided that the income level of less than 1000 US$ is the poor, income level of US$ 1000-1700 is the medium, and income level of more than 1700 US$ is the high level. 

Point 5: The OLS regression results are also confusing and hard to follow. Where the variable of way of accessing mangrove product came from? It is not introduced anywhere (not in Table 2) before. What does "native village" represent? What are non-farm and farm activities and how these variables are constructed i.e. what do they represent? Both non-farm and farm activities variables have negative signs, what do they mean? And, even if the model has 9 variables, R-squared is low (just .35), any comments/thoughts?

Response 5: I edited table (2). In this study, most respondents were heavily dependent on the forest regardless of the gender of the head of household, a similar to the finding of a study conducted by Abu Nasar Mohammad Abdullah [38]. Similarly, it has a negatively significant correlation with mangrove forest income. So, if the households have other alternative livelihood sources, they will be less dependent on mangrove forest products. So, a negative sign for non-farm activities and farm activities are expected signs. The model has 9 variables but 4 variables: gender, age, education and native village are not significantly. The community in that area heavily rely on mangroves but that is just subsistence level. They do not have regular income. And currently deforestation rate of mangrove in that area are higher than if it is compared with other area. So, only five variables are significantly significant and explained 35% of the variance on mangrove forest income.

 Point 6: What was the sampling frame of the study?

 Response 6: We added the sampling frame of the study. 

Point 7: The agricultural land size has a positive effect on mangrove forest income, which I think is not an expected sign. The positive sign means the higher the agricultural land size, the higher the forest income. How can it be explained? I don't know how mangrove forest and agriculture are linked in the study regions, but mangrove forest income is all about seafood and firewood.

 Response 7: Agricultural land size is positively correlated with mangrove forest income. This result is contradictory to the general findings of other studies Lebmeister et al., [39] observed that NTFPs dependency in the rural household was significantly decreased with increasing farmland. In Ethiopia, the relative income from the forest was negatively correlated with cropland [28]. In parts of the Ayeyarwady Delta, land degradation and declining soil fertility due to exploitative farming practices have contributed to decreasing agricultural yields. As a result, in order to maintain agricultural incomes and food production, farmers have resorted to cultivating even more land [40]. For instance, in coastal areas, converting mangrove areas to rice farms has resulted in seawater encroachment and salinization of soils, providing a source of income for only a short period of time before yields drop below economic levels [40].

Round 2

Reviewer 2 Report

The authors incorporated the suggested changes, but the language in the paper really needs further improvement.

Author Response

Dear Sir,

Thank you for your comments.

Your major and minor comments are previous to be a good quality paper.

We tried to improve the language editing.
